DISCOVERY REPORT

# Multiple light signaling pathways control solar tracking in sunflowers

**Christopher J. Brooks[1], Hagop S. Atamian[1,2], Stacey L. Harmer[1]***

1 Department of Plant Biology, University of California, Davis, Davis, California, United States of America,
2 Schmid College of Science and Technology, Chapman University, Orange, California, United States of America

* slharmer@ucdavis.edu

**Data Availability Statement:** All sequencing data have been deposited in the Gene Expression

## Abstract

Sunflowers are famous for their ability to track the sun throughout the day and then reorient at night to face east the following morning. This occurs by differential growth patterns, with the east sides of stems growing more during the day and the west sides of stems growing more at night. This process, termed heliotropism, is generally believed to be a specialized form of phototropism; however, the underlying mechanism is unknown. To better understand heliotropism, we compared gene expression patterns in plants undergoing phototropism in a controlled environment and in plants initiating and maintaining heliotropic growth in the field. We found the expected transcriptome signatures of phototropin-mediated phototropism in sunflower stems bending towards monochromatic blue light. Surprisingly, the expression patterns of these phototropism-regulated genes are quite different in heliotropic plants. Most genes rapidly induced during phototropism display only minor differences in expression across solar tracking stems. However, some genes that are both rapidly induced during phototropism and are implicated in growth responses to foliar shade are rapidly induced on the west sides of stems at the onset of heliotropism, suggesting a possible role for red light photoreceptors in solar tracking. To test the involvement of different photoreceptor signaling pathways in heliotropism, we modulated the light environment of plants initiating solar tracking. We found that depletion of either red and far-red light or blue light did not hinder the initiation or maintenance of heliotropism in the field. Together, our results suggest that the transcriptional regulation of heliotropism is distinct from phototropin-mediated phototropism and likely involves inputs from multiple light signaling pathways.

## Introduction

Plants display an extraordinary ability to alter their growth in response to environmental cues, a phenomenon that has fascinated scientists for hundreds of years. With light being a necessary resource, plants use alterations in light quality, intensity, and direction as cues to direct different growth behaviors [1,2]. This is accomplished using a wide range of photoreceptors, which sense these changes in light and then signal appropriate growth responses. For example, phytochromes are red/far-red photoreceptors that allow plants to sense neighbors and alter growth in an attempt to outgrow competitors in a process called shade avoidance [3]. Plants can sense

Omnibus database (GEO) (http://www.ncbi.nlm. nih.gov/geo) with accession number GSE229654. All other data are within the paper and its Supporting information files.

**Funding:** This work was supported by grants from the National Science Foundation (1238040 and 1759942) and the National Institute of Food and Agriculture (CA-D-PLB-2259-H) to SLH. The funders had no role in study design, data collection and analysis, decision to publish, or preparation of the manuscript.

**Competing interests:** The authors have declared that no competing interests exist.

**Abbreviations:** ARF, auxin response factor; *BIM1*, *BES1-INTERACTING MYC-LIKE 1*; GO, gene ontology; *HB2, HOMEOBOX PROTEIN 2*; *IAA4*, *INDOLE-3-ACETIC ACID INDUCIBLE4*; *NPH3*, *NON-PHOTOTROPIC HYPOCOTYL 3*; PCA, principal component analysis; *PHOT1*, *PHOTOTROPIN 1*; *PHOT2*, *PHOTOTROPIN 2*; *PHYB*, *PHYTOCHROME B*; *PIN7*, *PIN-FORMED7*; PP2C-D, D-CLADE TYPE 2C PROTEIN PHOSPHATASE; qRT-PCR, quantitative reverse transcription polymerase chain reaction; *SAUR*, *SMALL AUXIN UP-REGULATED*; *SAUR12*, *SMALL AUXIN UP-REGULATED RNA12*; UVR8, UV RESISTANCE LOCUS8; ZT, Zeitgeber Time.

UV light with UV RESISTANCE LOCUS8 (UVR8) and blue light with cryptochromes and phototropins [4–6]. These photoreceptors also affect plant form, with UVR8 and cryptochromes playing important roles in photomorphogenesis and the phototropins mediating directional growth towards blue light [7]. Finally, ZEITLUPE is a blue light photoreceptor that conveys information about the light environment to the plant circadian clock [8].

One of the best characterized growth responses in plants is phototropism. During this process, plants grow or bend either towards or away from a directional light source [9]. Research has for decades focused on the mechanisms regulating the phototropic bending of embryonic organs such as hypocotyls towards directional blue light [7]. Work primarily conducted on the *Arabidopsis thaliana* hypocotyl has revealed that the main photoreceptors mediating this response are 2 related membrane-associated protein kinases, *PHOTOTROPIN 1* (*PHOT1*) and *PHOTOTROPIN 2* (*PHOT2*). When exposed to blue light, interactions between phot1 and phot2 and another membrane-associated protein NON-PHOTOTROPIC HYPOCOTYL 3 (NPH3) are disrupted [10]. Although the signaling pathway is not fully understood, the phototropins and NPH3 are essential for the redistribution of auxin, which is carried out by the polar subcellular redistribution of *PIN-FORMED* (*PIN*) auxin transport proteins [11,12]. Auxin redistribution creates an auxin gradient within the hypocotyl, leading to higher auxin levels on the shaded sides of plants compared to the lit sides. In *Brassica oleracea* hypocotyls, phototropism-induced auxin gradients lead to an induction of multiple genes on the shaded sides of the stems. This up-regulation occurred prior to the growth response, suggesting a role for these auxin-induced genes in phototropic growth [13]. These genes include members of the *SMALL AUXIN UP-REGULATED* (*SAUR*) gene family. A subfamily of these auxin-induced SAUR proteins has been reported to inhibit D-CLADE TYPE 2C PROTEIN PHOSPHATASEs (PP2C-D), which, in turn, promote the phosphorylation and activation of plasma membrane proton ATPases. The subsequent extrusion of protons from cells leads to the acidification of the apoplast and cell expansion [14].

While responding to the environment is critical, plants also respond to internal cues to modulate growth. Plants exhibit autotropic control of posture so as to maintain an erect growth habit [15]. In this process, often called proprioception or autostraightening, portions of a stem bent during a tropic response subsequently straighten in a gravity-independent process [16–18]. The mechanisms underlying this proprioceptive response are still poorly understood; however, it is believed that there is a mechanical sensing component involving the actin–myosin XI cytoskeleton [19]. It has been proposed that specialized fiber cells with large bundles of actin filaments may act as propriocytes, sensing and/or responding to the stem bending [15,19]. While molecular details remain to be determined, there is evidence that, unlike phototropism, this process does not depend upon an auxin gradient across the straightening stem [20,21].

The circadian system has been shown to modulate plant responsiveness to a wide range of environmental cues, including tropic responses to light [22–24]. A functional circadian clock promotes plant fitness by coordinating internal processes with changes in the external environment [25]. The plant circadian oscillator is made up of a complex network of transcriptional regulators that control each other's expression and expression of thousands of output genes [26]. The circadian clock can generate daily growth patterns, with many plants displaying 24-hour rhythms in stem elongation and leaf movement even when maintained in constant environmental conditions [27].

One growth process that may incorporate responses to light, autostraightening, and circadian regulation is sunflower heliotropism. Sunflowers have an amazing ability to track the sun throughout the day and reorient to face east in anticipation of sunrise the following morning [22]. This solar tracking is accomplished by antiphasic growth patterns on the east and west sides

of the stem, with the east side growing more during the day and the west side growing more at night. Sunflowers moved from the field into constant environmental conditions with a fixed overhead light will continue oscillatory east–west stem bending movements for several days, suggesting a role for the circadian clock in solar tracking [22]. Moreover, these tracking movements can be recapitulated in a growth chamber using a simulated moving blue light source, suggesting a role for phototropin-mediated phototropism. Limited gene expression analysis revealed that several homologs of genes involved in auxin signaling are differentially expressed across the east and west sides of heliotropic stems, further suggesting a role for the well-characterized phototropin pathway in this process [22]. Although the above results suggest heliotropism may be a specialized form of phototropism, this has not previously been rigorously examined.

In the current study, we use RNA sequencing to define and compare the transcriptional profiles of sunflowers undergoing heliotropism, phototropism, and autostraightening to better understand relationships between these processes. We find that genes differentially expressed across stems bending towards directional blue light are enriched for auxin-related functions, consistent with previous findings in other species and organs [13]. However, we find limited overlap between genes and pathways differentially expressed across phototropic and autostraightening stems, consistent with distinct mechanisms regulating these responses. To our surprise, our analysis of genes differentially expressed across solar tracking stems also revealed limited similarity between the transcriptomic signatures of blue light–mediated phototropism and heliotropism. This prompted us to examine gene expression across the stems of plants just initiating solar tracking. Interestingly, genes differentially expressed during the onset of heliotropism are distinct both from those differentially expressed in established solar trackers and in plants bending towards blue light in a growth chamber. Instead, we find that genes implicated in shade avoidance responses have rapidly increased expression on the west sides of stems during the onset of heliotropism, suggesting a possible role for phytochrome photoreceptors in this process. However, depletion of red and far-red or of blue light had little effect on the initiation or maintenance of solar tracking in the field. Collectively, our data suggest that heliotropism is transcriptionally regulated by mechanisms distinct from classic phototropin-mediated phototropism and is likely controlled by multiple light signaling pathways.

## Results

### Little overlap between transcripts differentially expressed on the growing sides of stems during phototropism and autostraightening

Heliotropic plants bend from east to west during the day then from west to east during the night. We previously demonstrated a role for the circadian clock in this process [22], but the involvement of other signaling pathways has not been extensively investigated. We wished to explore the possibility that daytime movements rely upon the phototropin signaling pathway and that nighttime reorientation involves an autostraightening response. Autostraightening, also known as autotropism, has been most widely studied in plant organs undergoing a gravitropic response [17,28]. To investigate whether sunflower stems responding to a light cue also show this behavior, we exposed 2-week-old sunflower plants grown in light/dark cycles to unidirectional blue light and monitored stem angle over many hours (Fig 1A). We found that while plants initially bend rapidly towards the light source, after around 10 hours of continuous illumination, they indeed bend away from the light in an autostraightening response (Fig 1B).

We previously showed that sunflower phototropism is gated by the circadian clock, such that the degree of bending towards a unidirectional blue light over a 4-hour period is dependent on the time of day the stimulus is given [22]. We therefore next tested whether the autostraightening response is also time-of-day dependent: We exposed 2-week-old chamber-

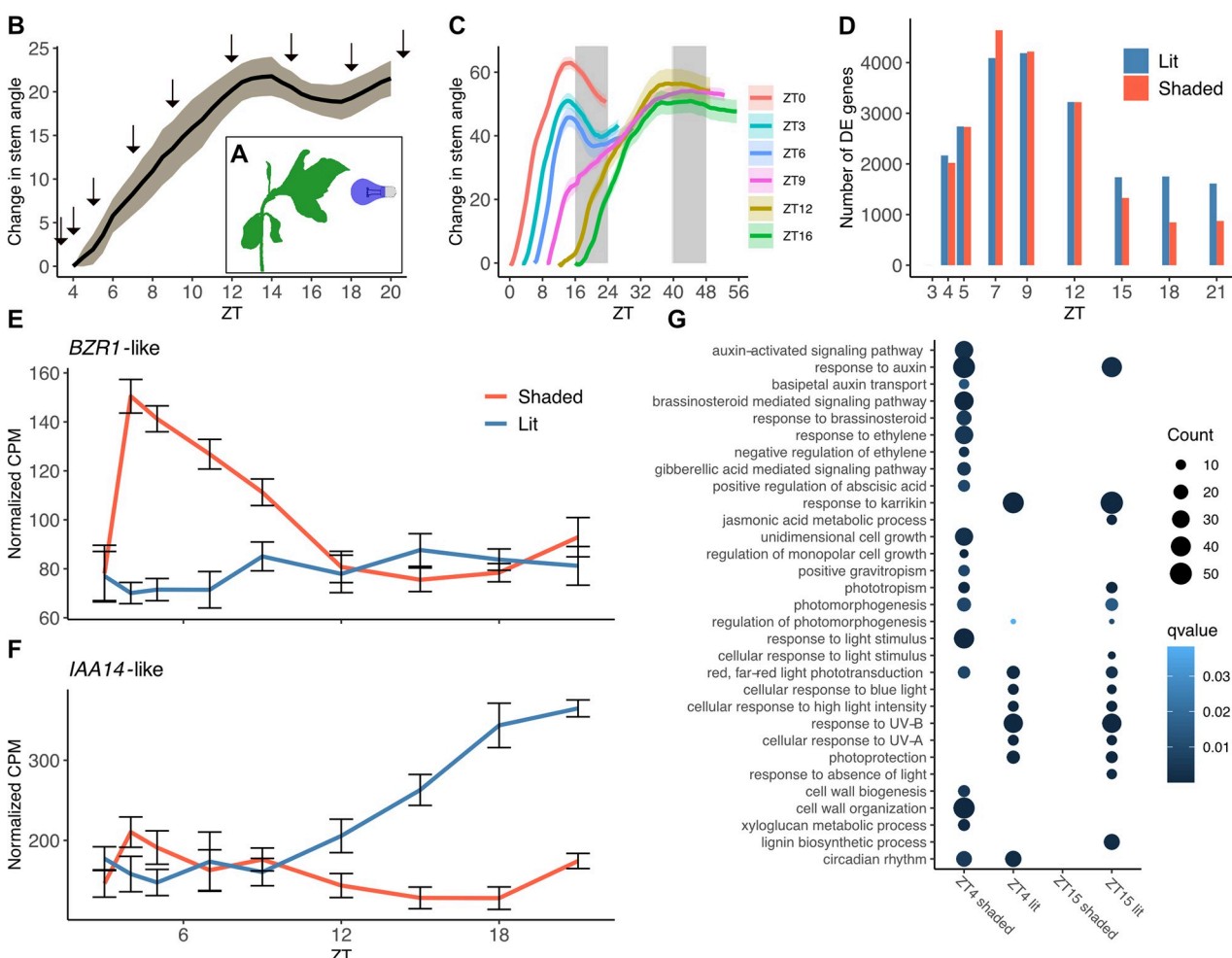

**Fig 1. Phototropic and autostraightening response in sunflower stems.** (A) Schematic of experimental design. (B) Changes in mean stem angles over time for 2-week old sunflower plants stimulated with unidirectional blue light at 3 hours after dawn ($n = 3$). Arrows indicate times of tissue collection for phototropism RNA-seq. (C) Change in mean stem angles over time for 2-week old sunflower plants stimulated with unidirectional blue light starting at different times of day ($n = 6$–9). White and gray areas represent subjective day and subjective night. Ribbons represent SEM. (D) Numbers of genes significantly more highly expressed on the lit (blue) or shaded (red) sides of stems are indicated. Significance determined using FDR < 0.05, based on pairwise comparisons of gene expression on the shaded and lit sides of the stem at indicated times. (E, F) Normalized CPM of *BZR1*-like (*Ha412HOChr14g0662481*) and *IAA14*-like (*Ha412HOChr03g0143141*) over time on shaded and lit sides of stems (mean +/− SEM, $n = 6$). (G) GO enrichment analysis of genes differentially expressed genes during phototropism at ZT4 and ZT15. All terms for processes involved in light signaling, hormone regulation, and growth are shown for lit and shaded sides of stems. The underlying raw data may be found in S1 Data for Fig 1D; in S2 Data for Fig 1G; in S13 Data for Fig 1B and 1C; and at NCBI GEO, accession GSE229654, for Fig 1E and 1F. CPM, counts per million; FDR, false discovery rate; GO, gene ontology; SEM, standard error of the mean; ZT, Zeitgeber Time.

grown sunflowers to unidirectional blue light at 6 different time points, ranging from the normal time of lights on (Zeitgeber Time 0, or ZT0) to lights off (ZT16) (Fig 1C). Sunflowers exposed to unidirectional blue light during the first part of the day (ZT0, ZT3, and ZT6) displayed a similar autostraightening response, in all cases beginning at around ZT14. However, plants exposed to unilateral light in the later part of the day (ZT9, ZT12, and ZT16) did not undergo autostraightening at ZT14, nor did they autostraighten when they reached a maximum bending angle many hours later. This suggests autostraightening is gated by the circadian clock, such that the timing of the onset of phototropic bending influences the ability of the plant stem to autostraighten.

We next wished to identify genes with expression patterns correlated with sunflower auto-straightening and phototropism. Many signaling components involved in phototropism have been characterized; however, most of this work has been done using the hypocotyls or coleoptiles of etiolated plants [7]. Moreover, little is known about the molecular mechanisms involved in autostraightening [17]. To identify candidate genes and pathways involved in sunflower stems undergoing phototropism and subsequent autostraightening, we performed RNA sequencing. Two-week-old light-grown plants were placed in front of unidirectional blue light at ZT3, a time permissive for robust phototropism and subsequent autostraightening. Stem peels, consisting of epidermal, cortex, and vascular tissue (S1 Fig), were collected from the lit and shaded sides of bending stems at 9 time points over an 18-hour period (Fig 1B). RNA was extracted, libraries constructed, and whole transcriptome sequencing was performed using the Illumina HiSeq platform. Reads were mapped to the Ha412HOv2 genome using the Eugene annotation [29]. Genes differentially expressed across the stem were determined by pairwise comparisons of gene expression between the lit and shaded sides at each time point. We found thousands of genes are differentially expressed across the 2 sides of stems, starting at 1 hour after exposure to the blue light and persisting until the end of our experiment (Fig 1D and S1 Data).

To implicate distinct biological processes in these bending responses, gene ontology (GO) enrichment analysis was performed on genes differentially expressed at each time point. As expected, genes rapidly up-regulated on the lit sides of stems were enriched for homologs of genes involved in blue light and UV responses (Fig 1G). As also expected based on previous studies of phototropism [12,13,30], homologs of auxin-responsive and auxin signaling genes were significantly enriched among transcripts quickly up-regulated on the shaded sides of stems (Figs 1G and S2 and S2 Data; note that sunflower gene IDs for every enriched GO term are listed in these files). The same was true for genes predicted to act in the signaling pathways of other growth-promoting hormones such as brassinosteroid and gibberellic acid (Fig 1E and 1G). Consistent with the onset of bending occurring within 1 hour after light stimulation (Fig 1B), homologs of genes involved in cell growth and cell wall processes were also enriched among those rapidly up-regulated on the shaded sides of stems. Overall, these data are consistent with previous studies suggesting that phototropic bending towards blue light is mediated by the redistribution of auxin towards the shaded sides of stems [12,31].

Interestingly, we also observed a highly significant enrichment of genes implicated in translational and ribosomal processes among those up-regulated on the shaded sides of stems between ZT7 and ZT15 (S3 Fig). The lag between the enrichment of these terms and both the onset of growth (Fig 1B) and the enrichment of hormonal and cell wall–related terms (S3 Fig) suggests that induction of the translation machinery occurs in response to growth rather than as its driver.

We next examined differential gene expression across stems during the autostraightening response, 12 hours after exposure to unidirectional blue light (ZT15) (Fig 1B). Since autostraightening is driven by the faster elongation of cells on the lit sides of stems and phototropism by the faster elongation of cells on the shaded sides of stems [16,32], we speculated that similar growth pathways might be involved in both processes. Indeed, we found homologs of auxin-responsive genes enriched among transcripts up-regulated on the lit sides of stems at ZT15, similar to the enrichment of these types of genes on the shaded sides of stems at ZT4 (Fig 1F and 1G). However, we did not find enrichment for homologs of genes involved in auxin signaling, or brassinosteroid or gibberellic acid signaling or response, among auto-straightening genes. Moreover, a direct comparison of genes differentially expressed at ZT4 and ZT15 revealed that few genes were up-regulated both on the shaded sides of stems at ZT4 and on the lit sides of stems at ZT15 (S4 Fig and S3 Data). The limited overlap between

enriched functional categories (Figs 1G and S2) and specific genes (S4 Fig) at the phototropic and autostraightening time points suggests that different growth pathways may control these different bending responses.

## The majority of genes expressed in heliotropic stems have diel rhythms with peak expression occurring either before dusk or dawn

Having identified genes implicated in phototropism and autostraightening, we next wanted to investigate patterns of gene expression in plants undergoing heliotropism in natural growth conditions (Fig 2A). We collected peels every 4 hours over 2 days from the east and west sides of 2-week-old sunflowers exhibiting robust solar tracking in the field (Fig 2B). RNA expression analysis was carried out as described for phototropic stems. To visualize the difference in expression profiles between samples, we performed principal component analysis (PCA) [33]. Both PC1 (22.2%) and PC2 (19.8%) separate the samples by time of day, suggesting that time of day rather than side of stem or date of collection is the most influential factor on gene expression in this experiment (Fig 2C).

To examine daily expression patterns, the Cosinor method available in the R package DiscoRhythm was used to estimate the acrophase, amplitude, and rhythmicity of each expressed gene [34,35]. Of the 34,071 expressed genes, 21,325 (62.6%) were assessed as having a diel rhythm (qvalue < 0.05) on both the east and west sides of the stem (Fig 2D and S4 Data). Although some genes appeared to be uniquely rhythmic on one side or the other, most of those are removed if more stringent statistical cutoffs for rhythmicity are applied, suggesting these may represent transcripts with marginal rhythmicity. Therefore, only the 21,325 genes scored as significantly rhythmic on both the east and west sides of stems were used for acrophase and amplitude analysis. These high-confidence rhythmic genes include homologs of core clock genes, *LHY-like* and *PRR5-like* [36], which displayed strong daily rhythms with similar peak phases to their Arabidopsis homologs [37,38] (Fig 2E and sF). Importantly, there was no significant difference in expression across the 2 sides of the stem either for these or other homologs of core clock genes (Figs 2E, 2F and S5). These data indicate that the circadian clock is robustly rhythmic and entrained to the same phase on the opposite sides of solar tracking stems.

We next examined the overall patterns of expression for genes with diel patterns of expression on the east and west sides of stems in field-grown sunflowers. Of these high-confidence rhythmic genes, the majority had peak expression either before dusk or before dawn (Fig 2G). To identify functional categories enriched among genes with peak expression at distinct times of day, we placed genes into 4-hour bins based on their times of peak expression and performed GO analysis (S5 Data). As expected, genes with peak expression in the early morning (ZT0 to ZT4) were enriched for terms involved in the response to different wavelengths of light and in photoprotection. Genes with peak expression in the first part of the day (ZT4 to ZT8) were enriched for heat response and protein folding, suggesting up-regulation of pathways involved in abiotic stress responses at this time. Genes with peak expression in the afternoon (ZT8 to ZT12) and in the evening (ZT12 to ZT16) were enriched for functions in DNA repair, DNA replication, and the cell cycle, suggesting cell division might be preferentially phased towards the end of the day. Intriguingly, we also found enrichment of genes implicated in RNA modification, especially in chloroplasts and mitochondria, in the afternoon and evening. RNA editing is essential for organelle biogenesis [39], suggesting that replication of endosymbiotic organelles may be coordinated with that of the nuclear genome. Homologs of brassinosteroid signaling components were enriched during the evening and early night (ZT12 to ZT16 and ZT16 to ZT20). This steroid hormone has been shown to play epidermal-

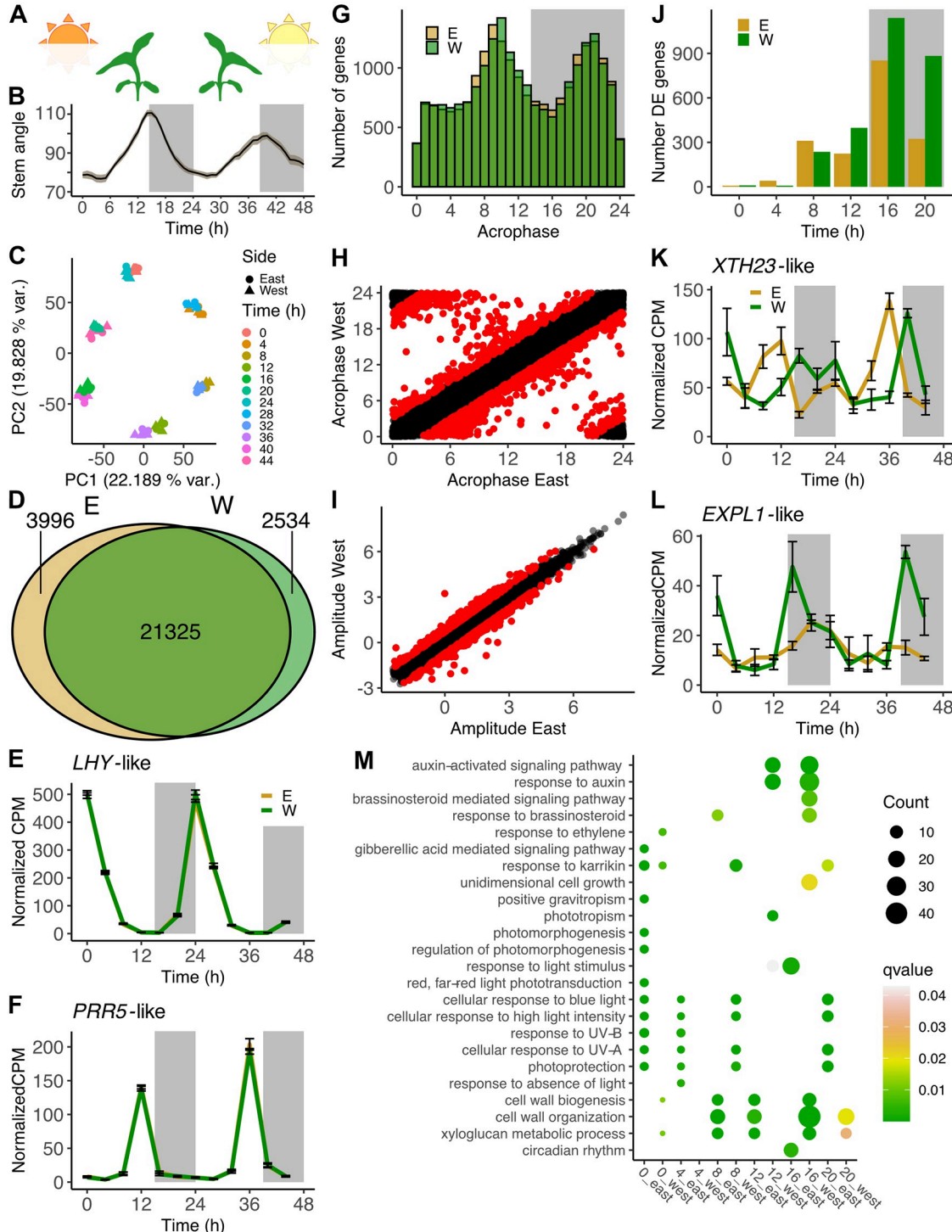

**Fig 2. Transcriptome analysis of solar tracking sunflower stems. (A)** Schematic showing orientation of heliotropic plants at dawn and dusk. **(B)** Stem angles of 2-week-old sunflowers growing in the field (mean plotted with ribbon representing SEM, *n* = 6). White and gray areas represent day and night. **(C)** PCA analysis of gene expression during heliotropism. **(D)** Number of genes with statistically significant diel rhythms on east or west sides of stems (FDR < 0.05). **(E, F)** Normalized expression over time for *LHY*-like (*Ha412HOChr15g0736201*) and *PRR5*-like (*Ha412HOChr08g0350221*) transcripts during heliotropism (mean +/− SEM, *n* = 3). **(G)** Acrophase distribution of genes that are significantly rhythmic on both sides of sunflower stems. **(H)** Acrophases of genes on the east and west sides of stems. Genes with an acrophase difference greater than 3 hours are highlighted in red. **(I)** Amplitudes of significantly rhythmic genes on the east side and west sides of stems. Genes with amplitude differences greater than 2-fold across the

stem are highlighted in red. **(J)** Numbers of genes significantly more highly expressed on the east or west sides are indicated (FDR < 0.05). Pairwise comparisons of gene expression on the east and west sides of stems at indicated times. **(K, L)** Normalized expression of *XTH23*-like and *EXPL1*-like transcripts during heliotropism (mean +/− SEM, *n* = 3). **(M)** GO enrichment analysis of genes differentially expressed during heliotropism. All terms shown in Fig 1G are included here for direct comparison. **(E, F, K, L)** Mean values are plotted +/− SEM, *n* = 3. The underlying raw data may be found in S4 Data for Fig 2D and 2G; in S6 Data for Fig 2H; in S7 Data for Fig 2I; in S8 Data for Fig 2J; in S9 Data for Fig 2M; in S13 Data for Fig 2B; and at NCBI GEO, accession GSE229654, for Fig 2C, 2E, 2F, 2K, and 2L. FDR, false discovery rate; GO, gene ontology; PCA, principal component analysis; SEM, standard error of the mean.

and vasculature-specific roles in local and systemic signaling [40–43]; it is possible that one or more of these processes is controlled in a time-of-day–dependent manner. Plant defense-related terms were enriched in the early night (ZT16 to ZT20), consistent with reports of reduced plant susceptibility to fungi and bacterial pathogens in the late night and at dawn compared to near dusk [44–46]. We also found enrichment for autophagy and cell growth terms in the early night (ZT16 to ZT20) and for auxin-related terms in the late night (ZT20 to ZT24). Overall, these data reveal that the majority of the stem transcriptome is rhythmically expressed in field-grown plants and suggest ways in which plant physiology may be temporally partitioned in natural conditions.

## Transcriptome analysis implicates auxin signaling in heliotropic growth at night but not during the day

We next wished to identify genes with different expression patterns on the east and west sides of heliotropic stems. For most genes, the diel expression patterns across the stem are quite similar, consistent with the near-identical patterns of expression of clock genes on the east and west sides of stems (Figs 2E, 2F and S5). However, when comparing the times of peak expression of transcripts significantly rhythmic on the 2 sides, we found that 1,369 genes have acrophase differences of greater than 3 hours on opposite sides of the stem (Fig 2H and S6 Data). Many of these, for example, *XTH23-like*, appear to have different waveforms on the 2 sides as well (Fig 2K). We next compared the rhythmic amplitude of transcripts on the east and west sides of stems (Fig 2I and S7 Data). As was true for our phase analysis, most genes were very similar, with only 811 having a greater than 2-fold difference in amplitude across the stem. Some of these transcripts, such as *EXPL1-like*, have much higher overall expression levels on one side of the stem compared to the other (Fig 2L).

We next compared expression across the stem at individual time points to identify genes and pathways associated with heliotropic movement. This analysis revealed relatively few differentially expressed genes early in the day, at ZT0 and ZT4, but a larger number of transcripts differentially expressed in the afternoon and early evening, ZT8 and ZT12. Surprisingly, we found the largest numbers of differentially expressed genes at night, ZT16 and ZT20 (Fig 2J and S8 Data), times when there cannot be unequal illumination across the stem. These data demonstrate there is significant differential expression across the stems of solar tracking plants, but not phased to the times of day we had expected.

We previously showed that the east sides of solar tracking stems grow faster than the west sides during the day while the west sides grow faster than the east sides at night [22]. To identify biological processes correlated with these growth patterns, we next performed GO analysis on genes differentially expressed at each time point. Since previous studies, along with our own findings, have suggested involvement of light responses, hormone signaling, and growth-related processes in phototropism [9], we focused on these terms. We found enrichment for blue light and UV responses on the east sides of stems in the early morning (ZT0 and ZT4), as

well on the west sides in the early afternoon (ZT8), consistent with the position of the sun relative to plant stems at these times (Fig 2A and 2M). We next considered enriched growth-related GO categories. As expected, we found enrichment for cell wall–related processes at ZT8 and ZT12 on the east side, and at ZT16 and ZT20 on the west side, coinciding with the times of antiphasic growth on the opposite sides of the stem (Fig 2B). Surprisingly, however, auxin signaling and response terms were not enriched among genes more highly expressed on the east side during the day. However, these terms were enriched among genes more highly expressed on the west side in the evening and early night (ZT12 and ZT16) (Fig 2M and S9 Data). These results suggest a potential role for the auxin signaling pathway during nighttime west-to-east reorientation, but not during daytime east-to-west growth.

## Transcriptional responses to phototropism and heliotropism are quite distinct

To investigate whether phototropism towards blue light and heliotropism in the field rely on similar molecular pathways, we next systematically compared the patterns of gene expression occurring during these 2 processes. First, we asked if genes with different diel phases across the stem during heliotropism (Fig 2H) might also be differentially expressed across the stem during phototropism. We examined the expression of *S15a-like*, which encodes a protein with homology to *Arabidopsis thaliana* ribosomal protein *S15a* (AT1G07770). During heliotropism, this gene has a phase difference of 4.2 hours between the east and west sides, with a peak expression later in the day on the east side of the stem compared to the west (Fig 3A). This gene is also differentially expressed in the chamber during phototropism, with increased levels on the shaded side of the stem between 4 and 12 hours after the onset of directional illumination (Fig 3B). To determine if this pattern is consistent for other genes with different phases of peak expression across heliotropic stems, we subdivided all differentially phased genes into 3 subcategories based on times of peak expression (Figs 3C and S6). Similar to *S15a-like*, the normalized average expression of all genes with comparable peak phases of expression in the field is higher on the shaded side of the stem 4 to 12 hours after the onset of directional illumination during phototropism (Fig 3D). Gene expression patterns during phototropism were similar for the 2 other groups of transcripts with different phases of expression across heliotropic stems (S6 Fig). In all cases, up-regulation on the shaded sides of stems occurs many hours after the initial phototropic bending (Fig 1B). GO analysis of transcripts in all 3 phase groups revealed enrichment for terms involved in ribosomal biogenesis, translation, and DNA replication (S10 Data). This enrichment suggests roles for these genes in cellular growth induced in response to bending, rather than for roles in the initiation of this tropic response. Overall, transcripts expressed with different phases across solar tracking stems are correlated neither with the rapid onset of growth on the shaded sides of stems nor with autostraightening growth on the lit sides of stems.

Given that hundreds of genes have different amplitudes of expression on the opposite sides of heliotropic stems (Fig 2I), we investigated their expression patterns during phototropism. The normalized mean expression patterns of these transcripts in phototropic plants reveals higher expression on the shaded sides of stems 4 to 12 hours after the onset of directional illumination during phototropism (S7 Fig), with up-regulation occurring much slower than the onset of plant bending (Fig 1B). GO analysis of these transcripts did not reveal any significantly enriched functional categories. Although the pattern of expression of these genes during phototropism is similar to that observed for transcripts with different phases of expression in solar tracking plants (Figs 3D and S6), there is only limited overlap between genes expressed with different phases and amplitudes in the field (S7 Fig). As is true for the differentially

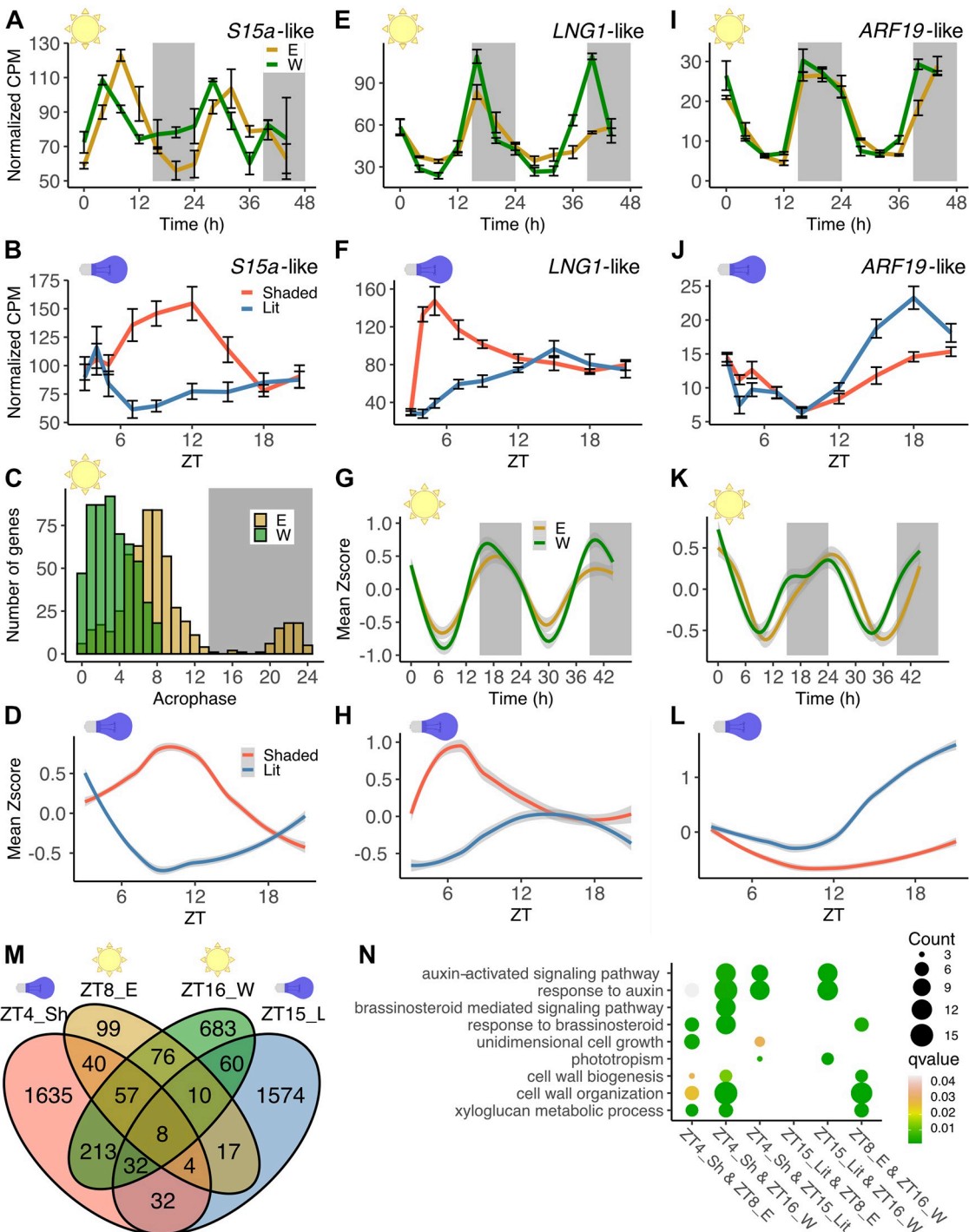

**Fig 3. Distinct patterns of gene expression during phototropism and heliotropism. (A, B)** Normalized transcript levels over time for *S15a*-like (*Ha412HOChr05g0204731*) during heliotropism and phototropism, respectively. **(C)** Genes with different acrophases on the east and west sides of stems; west side acrophases between ZT0 and ZT8. **(D)** Zscore-normalized mean expression of the differentially phased genes shown in (**C**) during phototropism. **(E, F)** Normalized transcript levels over time for *LNG1*-like (*Ha412HOChr17g0822481*) during heliotropism and phototropism, respectively. **(G, H)** Expression patterns of phototropic genes. Their Zscore-normalized mean expression during heliotropism and phototropism, respectively, are shown. **(I, J)** Normalized transcript levels over time for *ARF19*-like (*Ha412HOChr03g0103791*) during heliotropism and phototropism, respectively. **(K, L)** Expression patterns of autostraightening genes. Their Zscore-normalized mean expression during heliotropism and phototropism, respectively, are shown. **(M)** Numbers of genes more highly expressed on the growing sides of stems during phototropism (ZT4 shaded (ZT4_Sh) and ZT15 lit (ZT15_L)) and during heliotropism (ZT8 east (ZT8_E) and ZT16 west (ZT16_W)). **(N)** GO

enrichment analysis of genes shared between any 2 of the growth categories shown in (**M**). Terms shown are those shared with Figs 1G and 2M. (**A, E, I**) Mean CPM +/− SEM, *n* = 3. (**B, F, J**) Mean CPM +/− SEM, *n* = 6. (**G, K**) Lines fitted with a generalized additive model. (**D, H, L**) Lines fitted with LOESS. Ribbons represent 95% confidence intervals. The underlying raw data may be found at NCBI GEO, accession GSE229654, for Fig 3A, 3B, and 3D-L; in S3 and S8 Data for Fig 3M; in S6 Data for Fig 3C; in S17 Data for Fig 3D, 3G, 3H, 3K, and 3L; and in S18 Data for Fig 3N. CPM, counts per million; GO, gene ontology; SEM, standard error of the mean; ZT, Zeitgeber Time.

phased genes, transcripts expressed with different amplitudes across solar tracking stems are not correlated with the rapid onset of growth on the shaded sides of stems or with autostraightening growth on the lit sides of stems.

Since we found little overlap between genes expressed with different rhythmic parameters on the opposite sides of heliotropic stems and genes differentially expressed during the onset of phototropic bending, we next examined how genes correlated with phototropic or autostraightening growth are expressed during heliotropism. One transcript rapidly up-regulated on the shaded sides of stems during phototropism is *LNG1-like* (Fig 3F), a homolog of the *Arabidopsis* gene *LONGIFOLIA1* (AT5G15580), which promotes unidimensional cell growth and is induced rapidly in response to auxin [47–49]. During heliotropism, this gene has a similar waveform and phase on the east and west sides of stems but a higher amplitude on the west side (Fig 3E). This results in a modestly higher level of expression on the east side of the stem around midday (ZT4 to ZT8), and on the west side of the stem in the late day and early night (ZT12 to ZT16). To determine whether this pattern is consistent for all genes rapidly up-regulated on the shaded sides of stems during phototropism, we examined their mean normalized expression in field-grown plants undergoing heliotropism. We found the same pattern of modest amplitude differences across the stem observed for *LNG1-like* (Fig 3G). This pattern is quite distinct from the large differences in mean expression levels and kinetics observed for these genes across stems during phototropism (Fig 3H). Overall, genes rapidly up-regulated on the shaded sides of stems during phototropism show only limited differences in expression across the stems of solar tracking plants.

We next reasoned that genes implicated in autostraightening responses might be differentially expressed during solar tracking. We first examined *ARF19-like*, a transcript that encodes a protein with homology to auxin response factors (ARFs) and that is a likely activator of transcription [50]. This ARF-like transcript is up-regulated late in the phototropic time course on the lit sides of stems (Fig 3J), correlating with the timing of the autostraightening response (Fig 1B). *ARF19-like* has very similar expression patterns on the east and west sides of heliotropic stems (Fig 3I). However, inspection of the mean expression patterns of all transcripts correlated with autostraightening (Fig 3L) during heliotropism revealed a slightly advanced late-afternoon/early-night phase on the west side of solar tracking stems compared to the east side (Fig 3K). This pattern can also be observed for the *ARF19-like* transcript on the second day of the solar tracking time course (Fig 3I). Therefore, similar to our findings for genes correlated with phototropism, genes with expression patterns correlated with autostraightening have only subtle differences in expression across the stems of solar tracking plants.

Because phototropic and autostraightening genes displayed slight differences in amplitude and phase on the opposite sides of plant stems during heliotropism, we wanted to directly compare genes scored as differentially expressed during phototropism, autostraightening, or heliotropism at times of asymmetrical growth. We compared genes: up-regulated on the shaded sides of phototropic stems at ZT4 (phototropic genes), up-regulated on the lit sides of phototropic stems at ZT15 (autostraightening genes), up-regulated on the east sides of field-grown plants at ZT8 (daytime heliotropism), and up-regulated on the west sides of field-

grown plants at ZT16 (nighttime heliotropism) (Fig 3M). Pairwise comparisons revealed small but statistically significant overlaps between all of these categories (S1 Table). To investigate possible biological functions of genes shared between 2 categories, we next performed GO analysis. Transcripts present in both the phototropic (ZT4, shaded sides of stems) and daytime heliotropic (ZT8, east sides of stems) categories are enriched for unidimensional cell growth, cell wall organization, and xyloglucan metabolic processes (Fig 3N). However, auxin-related terms are not enriched among genes differentially expressed in both conditions. In contrast, transcripts differentially expressed in both the phototropic (ZT4, shaded sides of stems) and nighttime heliotropic (ZT16, west sides of stems) conditions are enriched for auxin response and auxin signaling terms. These auxin terms are also enriched among the transcripts differentially expressed in both the autostraightening (ZT15, lit sides of stems) and nighttime heliotropic (ZT16, west sides of stems) conditions. These data suggest that although there is significant overlap between transcripts differentially expressed during heliotropism and phototropism, those shared between phototropic and daytime heliotropic growth are more likely involved in carrying out growth programs. In contrast, transcripts shared between phototropic, autostraightening, and nighttime heliotropic reorientation are more likely responsible for initiating growth.

## Gene expression analysis suggests the onset of heliotropism and blue light–dependent phototropism are regulated by distinct pathways

We next wondered if the limited similarity between genes differentially expressed during daytime heliotropism and in phototropism was because the latter process is an acute response to light while the former involves the circadian system [22]. While sunflowers grown in a chamber with overhead light do not bend [51], heliotropic-type movements can quickly be induced using directional lights sequentially turned on and off to mimic the movement of the sun [22]. In addition, rhythmic bending can be observed for several days after plants are moved from the field into constant environmental conditions. Thus, while directional light is necessary to initiate tracking, and later to reinforce it, it is not absolutely required for heliotropic growth patterns. We therefore asked whether the onset and continuation of solar tracking are characterized by different transcriptional programs and in particular whether the initial onset is transcriptionally similar to phototropism in a growth chamber. We therefore characterized the growth and gene expression patterns of plants at the onset of heliotropism. To this end, 2-week-old chamber-grown plants were transplanted into the field after dusk and imaged for 3 days. During the first day in the field, sunflowers bent east a small amount in the morning and bent west to a greater degree in the afternoon. To our surprise, these plants reoriented during the first night so that they were inclined to the east before dawn (Fig 4A). These data suggest that heliotropism begins the first day young plants are transferred to the field.

Next, we examined transcriptional responses across the stem during these first few days. Two-week-old chamber-grown sunflowers were transplanted into the field as described above, and peels from the east and west sides of the stem were collected every 2 hours during the first day and every 4 hours for 2 more days. RNA was extracted from the stem peels, and gene expression was analyzed by quantitative reverse transcription polymerase chain reaction (qRT-PCR). First, we examined expression of genes that were rapidly up-regulated and then quickly down-regulated on the shaded sides of stems during phototropism (Figs 3H and S8). These included homologs of the known Arabidopsis auxin-related genes *INDOLE-3-ACETIC ACID INDUCIBLE4* (*IAA4*), *SMALL AUXIN UP-REGULATED RNA12* (*SAUR12*), and *PIN-FORMED7* (*PIN7*) [52]. Surprisingly, only one of these auxin-related transcripts (*IAA4*-like (*Ha412HOChr10g0435441*)) was differentially expressed across the stem at any time point

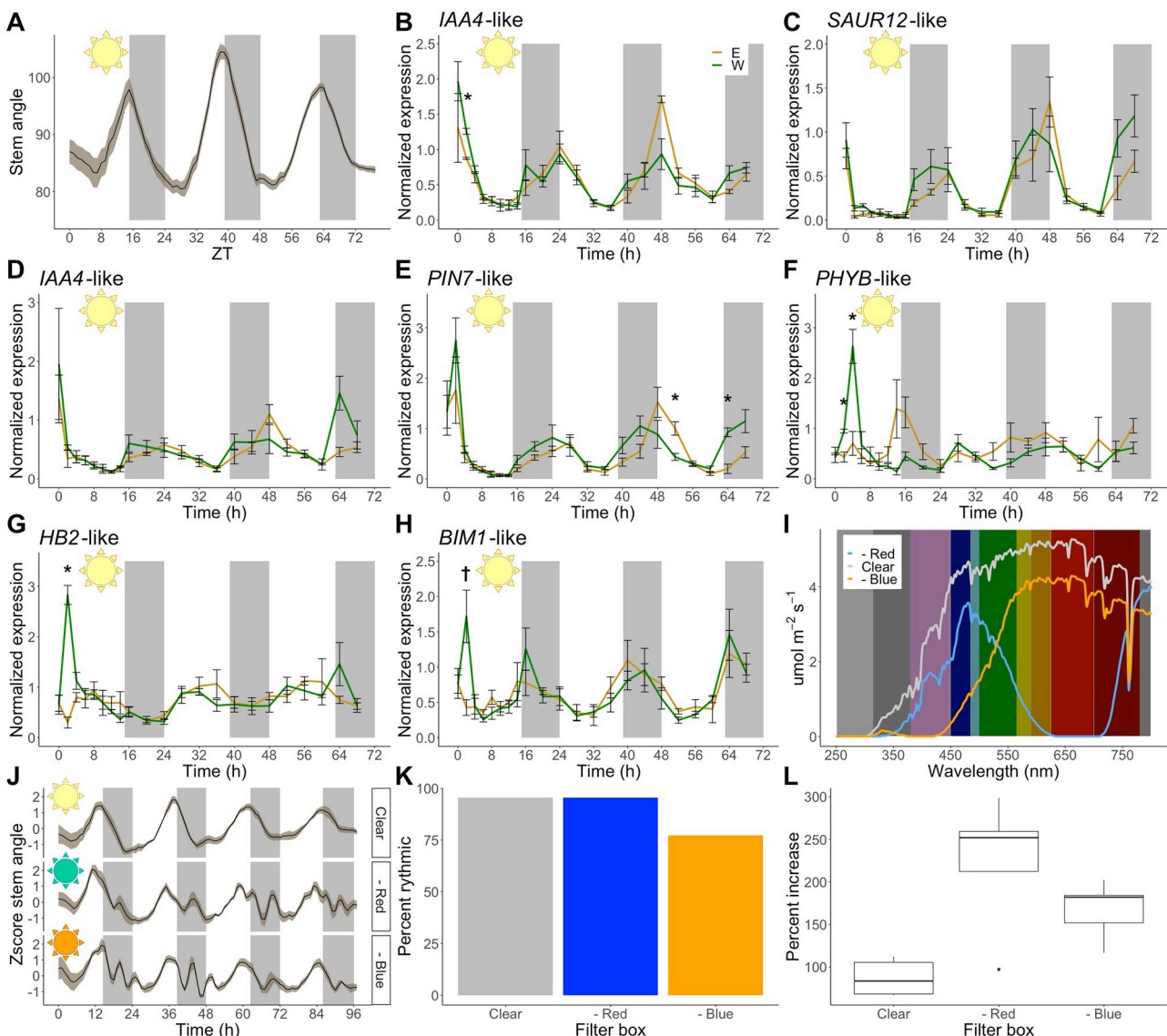

**Fig 4. Initiation of heliotropism shows a distinct transcriptional profile and occurs in different light conditions. (A)** Stem angles of 2-week old chamber-grown sunflowers transplanted into the field after sunset (mean +/− SEM, $n = 17$). **(B-H)** Normalized expression of selected phototropically regulated genes during the first 3 days in the field (mean +/− SEM, $n = 3$, except ZT64 west, $n = 2$). Statistical significance calculated using Welch's $t$ test, * represents qvalue < 0.05; † represents qvalue = 0.063. **(B)** *IAA4*-like (*Ha412HOChr10g0435441*). **(C)** *SAUR12*-like (*Ha412HOChr07g0303111*). **(D)** *IAA4*-like (*Ha412HOChr01g0004341*). **(E)** *PIN7*-like (*Ha412HOChr01g0035481*). **(F)** *PHYB*-like (*Ha412HOChr02g0079361*). **(G)** *HB2*-like (*Ha412HOChr13g0623611*). **(H)** *BIM1*-like (*Ha412HOChr02g0064141*). **(I)** Wavelengths of light measured inside 3 filter boxes. **(J)** Zscore-normalized stem angles of sunflowers inside filter boxes after their transfer from growth chamber to field (mean +/− SEM, $n = 5$). Data from one representative experiment. **(K)** Percentage of plants with significant rhythmic solar tracking in each of the 3 filter boxes (pvalue < 0.001, $n = 22$). **(L)** Percent increase in length of sunflower stems in each filter box **(J)** from dawn on day 1 to dawn on day 5. The underlying raw data may be found in S13 Data for Fig 4J; in S14 Data for Fig 4A-H; and in S15 Data for Fig 4I, 4K, and 4L. *BIM1, BES1-INTERACTING MYC-LIKE1; HB2, HOMEOBOX PROTEIN 2; IAA4, INDOLE-3-ACETIC ACID INDUCIBLE4; PHYB, PHYTOCHROME B; PIN7, PIN-FORMED7; SAUR12, SMALL AUXIN UP-REGULATED RNA12;* SEM, standard error of the mean; ZT, Zeitgeber Time.

during the first day in the field despite the obvious heliotropic bending during that time (Figs 4B–4E and S8A–S8D). However, at no time did we observe the rapid and one-sided up-regulation of expression these transcripts demonstrate during phototropism in the growth chamber (S8A–S8D Fig). During the second and third days, however, their expression patterns began to

mirror those observed in the RNA-seq data generated from plants grown in the field for over a week (S8A–S8D Fig).

We next examined the expression of 2 transcripts that undergo rapid and sustained up-regulation on the shaded sides of stems during phototropism (S8E and S8F Fig). These encode homologs of Arabidopsis *PHYTOCHROME B (PHYB)* and *HOMEOBOX PROTEIN 2 (HB2)*, genes known to be involved in shade avoidance [53,54]. These transcripts are rapidly up-regulated on the west sides of stems 2 hours after dawn; however, this is only true on the first day after transfer to the field (Fig 4F and 4G). A homolog of *BES1-INTERACTING MYC-LIKE1 (BIM1)*, another gene involved in shade avoidance [55], also had increased expression on the west side at ZT2, albeit not at levels reaching statistical significance (Fig 4H). We next examined the expression patterns of these 3 genes in our RNA-seq data of plants undergoing established solar tracking. Only the homolog of *HB2* is differentially expressed in this condition, being more highly expressed on the east sides of stems at ZT8 (S8 Data). In contrast to what we observed in plants initiating heliotropism, none of these 3 genes are similarly differentially expressed across the stems of established heliotropic sunflowers early in the day. Together, these data demonstrate that even though plants responding to unidirectional blue light in a chamber and those responding to the rising sun on their first day in the field exhibit similar bending, their underlying transcriptional responses are quite different. Our data also suggest the potential involvement of phytochrome-mediated signaling pathways in the initiation of solar tracking.

## Sunflower solar tracking is not affected by alterations in light quality

Our gene expression analysis suggested that early transcriptional events in heliotropism might have more in common with shade avoidance responses than with classical blue light–mediated phototropism. To investigate the possible involvement of multiple light signaling pathways in heliotropism, we tested the effects of changing light quality on the onset and maintenance of solar tracking. To do this, we built boxes using 3 types of filters: one clear, one to limit blue light, and one to limit red light (S9 Fig). Light measurements were taken from inside each box (Fig 4I) in the field. Light inside the clear box had a very similar spectrum to that of unfiltered light measured in the field and was used as a control. In the blue-depleted box, blue as well as violet and UV wavelengths were strongly reduced, while the red-depleted box removed virtually all red and a substantial portion of far-red light (Fig 4I). Soon after dusk, we transplanted 2-week-old chamber-grown sunflowers into one of these 3 boxes in the field and acquired time-lapse images of plant bending. Measurement of stem angles revealed that plants in all 3 types of boxes began tracking the first day, followed by reorientation on the first night (Fig 4J). Plants in all conditions maintained tracking movements for the following 3 days. However, plants in both the blue-depleted and red-depleted boxes exhibited additional movements at night (Fig 4J). The stems of these plants grew much longer than those in the clear box, likely due both to light quality differences and to overall lower light levels in these boxes compared to the clear box (Fig 4L). We suspect the noise in the nighttime reorientation of the plants grown in these 2 depletion boxes is caused by increased circumnutation accompanying the greater growth rates in these conditions compared to the clear box [56]. Despite these increased movements at night, estimations of rhythmicity of the bending movements revealed that over 75% of the plants in each condition, over 5 separate experiments, had significantly rhythmic tracking movements (Fig 4K). Of the 3 conditions, the blue-depleted plants had the lowest percentage of rhythmic trackers, perhaps due to the strong biphasic growth pattern, but overall diel rhythmicity of bending is still very robust (Fig 4J). These data show that both the onset and maintenance of heliotropism are resistant to changes in light

quality and suggest that multiple light signaling pathways are involved in the regulation of solar tracking.

## Discussion

Prior to anthesis, the stems of sunflower plants elongate in an antiphasic manner such that the east half grows more during the day and the west half grows more at night [22]. This allows plants to track the sun throughout the day and reorient to face east the following morning. We and others have previously assumed that heliotropism is a specialized form of phototropism [22,57–59]. However, here, we report that the transcriptional profile of solar tracking plants is quite different from that of plants responding to directional blue light in a growth chamber. We also show that solar tracking can begin on the first day plants are transplanted to the field and that the transcriptional response during the onset of heliotropism is distinct both from that seen during phototropism and in established solar trackers. Surprisingly, homologs of genes known to be involved in shade avoidance in Arabidopsis have increased expression on the west side of sunflower stems during the first few hours of their first day in the field, suggesting a potential role for phytochromes in the onset of heliotropism. Additional support for multiple light signaling pathways in solar tracking is provided by the robust tracking rhythms seen during both the onset and the maintenance of heliotropism in plants grown in red- or blue-depleted light conditions.

The role of auxin in phototropism has been best characterized in etiolated hypocotyls and coleoptiles, where higher levels of auxin accumulate on the shaded sides of stems [9]. These higher auxin levels lead to increased expression of auxin-regulated and growth-promoting genes preceding plant curvature towards light [13]. However, genetic studies have revealed different signaling mechanisms function downstream of phototropins in etiolated and de-etiolated Arabidopsis seedlings [60]. Our RNA-seq analysis of transcriptional responses across green phototropic sunflower stems revealed increased expression of thousands of genes on the shaded sides of stems within an hour of directional illumination. These rapidly induced genes are enriched for homologs of auxin-regulated and auxin-signaling genes as well as of other hormone-regulated and growth-promoting genes, similar to those found in phototropic etiolated *Brassica oleraceae* hypocotyls (Fig 1G; [13]). Our data therefore suggest that auxin relocalization plays an important role in the phototropic bending of de-etiolated sunflower stems towards blue light (S10A Fig), consistent with the well-established models for phototropism established primarily in etiolated seedlings [7,9].

The length of our phototropism time course also allowed us to analyze the transcriptional changes occurring later, when sunflower plants begin bending away from the light source in an autostraightening response. To our knowledge, this is the first report of genome-wide changes in gene expression during a proprioceptive response. We find thousands of genes are more highly expressed on the lit than the shaded sides of phototropically bending stems at the time when the plants begin to straighten (Fig 1D and S1 Data). Along with the expected higher expression of homologs of light-regulated genes, these genes are also enriched for homologs of auxin-responsive genes (Fig 1G). This suggests that autostraightening may be mediated by auxin, analogous to the auxin-mediated control of phototropism and gravitropism [7,20,61]. However, studies in pea epicotyls autostraightening after a gravitropic stimulus revealed that the reversal in growth asymmetry during this process is not accompanied by a reversal of the auxin gradient [20]. This conclusion was supported by studies in etiolated Arabidopsis seedlings expressing an auxin-responsive reporter gene [21]. Therefore, the relationship between auxin and autostraightening remains mysterious.

The circadian clock regulates transcription of a large fraction of plant genes, such that hundreds of genes can be significantly differentially expressed when the same genotype is sampled at even 30-minute intervals [62]. Our field RNA-seq data highlight the importance of time-of-day in the control of gene expression, as the time of collection has a much larger impact on gene expression than the side of stem or date of collection (Fig 2C). We found 62% of sunflower genes have robust diel rhythms, considerably more than the roughly 1/3 of expressed transcripts found to cycle in constant environmental conditions in Arabidopsis and sugarcane [63,64]. However, in sugarcane maintained in field conditions with cycling light, temperature, and humidity, 68% of genes in leaf tissue are rhythmic [65], similar to our results in sunflower. Notably, a significant proportion of rhythmic sunflower transcripts peak either in the late afternoon (ZT10) or in the late night (ZT22), in anticipation of the environmental transitions of dusk and dawn (Fig 2G). Similarly, bimodal phase distributions of rhythmic gene expression have been observed in other plants and even in diurnal primates [66–68]. These data suggest important roles for both the circadian clock and environmental response pathways in the control of gene expression in the natural environment.

Although time-of-day is the predominant determinant of differences in gene expression in our field experiment (Fig 2C), we identified many genes differentially expressed between the east and west sides of solar tracking sunflower stems. We were surprised to find relatively few genes differentially expressed across stems in the morning while thousands are differentially expressed in the late day and night (Fig 2J). The time point with the most differential expression across the stem is in the early night (ZT16), a time when there cannot be a light gradient across stems but when the west sides of stems are growing faster than the east sides [22]. The other nighttime time point, ZT20, has the second largest number of differentially expressed genes across the stem. Together, more than 70% of the differential expression detected in our heliotropism experiment occurs during the night (Fig 2J and S8 Data), even though these samples only represent 1/3 of our time course. This suggests that internal cues, rather than direct responses to environmental signals, predominantly control heliotropic growth patterns.

We and others previously suggested that auxin relocalization plays an important role in the bending of stems from east to west during the day [22]. However, we did not observe any enrichment of auxin-related GO terms among genes with higher expression on the east sides of stems during the day (Fig 2M). This is in marked contrast to the enrichment of auxin terms among genes up-regulated on the shaded sides of stems of sunflowers bending towards blue light in a growth chamber (Fig 1G). These data call into question the presumption that daytime solar tracking movements are driven primarily by the canonical phototropin signaling pathway. On the other hand, genes with higher expression on the west sides of stems at night are enriched for predicted functions in auxin signaling and response (Fig 2M). Together, this suggests that auxin may play a role in nighttime, but perhaps not daytime, reorientation of solar tracking stems (S10B Fig).

Direct comparisons of gene expression during phototropism and solar tracking highlight additional differences between these processes. The genes most differentially expressed across the stems of heliotropic plants have large differences in amplitudes or phases of expression (Fig 2H and 2I). While these 2 groups of genes are also generally differentially expressed in phototropic plants, their expression kinetics and GO term enrichment (Figs 3D, 3N and S6) suggest that they act in cellular maintenance pathways induced in response to elongation growth rather than acting to initiate bending. Similarly, transcripts with expression patterns correlated with the onset of phototropic growth in the growth chamber show only modest differences in expression across heliotropic stems (Fig 3G and 3H). The lack of morning induction of phototropism- or auxin-related genes on the east sides of solar tracking sunflowers

suggests phototropin-mediated auxin relocalization may not be the basis for daytime bending movements of heliotropic plants (S10B Fig).

We considered the possibility that the initiation of solar tracking and its maintenance might be mediated by different pathways. For example, Arabidopsis seedlings exposed to long periods of shade undergo rewiring of the connections between light and auxin signaling components such that sustained growth becomes independent of enhanced auxin levels [69]. We tested this by examining the expression patterns of 9 transcripts rapidly and transiently induced by unilateral blue light in the growth chamber in plants initiating solar tracking in the field. None of these transcripts are differentially expressed across stems of these field-grown plants during the first day, despite the rapid induction of daytime bending movements (Figs 4A–4E and S8). However, several transcripts whose homologs are involved in shade avoidance responses in Arabidopsis are rapidly induced on the west sides of stems in plants just moved to the field (Figs 4F–4H and S8). After a full day outside, however, these transcripts have very similar expression patterns on the east and west sides of solar tracking stems. These data suggest that similar to shade avoidance in Arabidopsis [69], the initiation and maintenance of heliotropism in sunflower may rely on different genetic networks.

The dissimilarity between gene expression in heliotropic plants in natural light and phototropic plants in monochromatic blue light leads us to suggest that multiple photoreceptors contribute to heliotropic movements in sunflower. This possibility is bolstered by the ability of plants to robustly initiate and maintain solar tracking movements in both blue-depleted and red-depleted light conditions (Fig 4I–4K). While abundant work has shown that phototropins are key mediators of hypocotyl bending towards blue light [7,9], other photoreceptors also regulate seedling phototropism. For example, in green Arabidopsis seedlings, phytochromes and cryptochromes have been reported to inhibit phototropism [60]. And while in most cases monochromatic red light has been reported to not induce phototropism, several papers have implicated phytochromes in positive phototropism of aerial tissues towards red light and away from far-red light [70–74]. Finally, a recent publication examining the bending of Arabidopsis inflorescence stems towards the sun showed a major role for cryptochromes, with only minor contributions from phototropins and UVR8 in this response [58]. Thus, we propose that in the natural environment, the initiation and maintenance of heliotropism in sunflower likely depends on both blue and red light photoreceptors in a process distinct from classical phototropin-mediated phototropism.

While in this study we have focused on transcriptional regulation, it is possible that post-translational regulatory mechanisms play a prominent role in the control of heliotropism. For example, it has been proposed that cytoskeletal components are involved in sensing and responding to stem curvature [19,75,76]. Stem bending in poplar has been reported to generate a propagated electrical signal, perhaps generated by mechanosensitive membrane channels [77,78]. Finally, while auxin can promote growth through transcriptional regulation, it has recently been reported that it can also rapidly activate kinases that directly phosphorylate and activate plasma membrane $H^+$-ATPases [79–81]. Further studies are required to determine whether any of the above processes contribute to the control of heliotropism.

## Materials and methods

### Plant growth conditions

Sunflower seeds (HA412-HO) were obtained from USDA North Central Regional Plant Introduction Station (USDA Germplasm Resources Information Network ID: PI 603993). Seeds were germinated in flats (filled with SunGro Sunshine mix #1) covered with a lid for 2 days. Lids were then removed and plants were grown in controlled environment chambers in 16

hours light:8 hours dark with 180 μmol m$^{-2}$ s$^{-1}$ fluorescent light at constant 25 ˚C. For phototropism experiments, 2-week old sunflowers were placed in front of unidirectional blue lights (Yescom 225 Blue LEDs Grow Light Ultrathin Panel, 280 μmol m$^{-2}$ s$^{-1}$) in a growth chamber at constant 25 ˚C. Sunflowers used for the onset of heliotropism experiments were grown in chambers with cycling temperature and photoperiod mimicking the field conditions for 2 weeks after germination and were then transplanted to the field after the sun had fully set. Plants used for established heliotropism field experiments were transplanted to the field 4 to 5 days after germination.

## Tropism measurements

For phototropic and autostraightening bending analysis, time-lapse images were taken with a raspberry Pi 3 with infrared light and camera adapted from (https://maker.danforthcenter.org/tutorial/raspberry%20pi/led/raspberry%20pi%20camera/RPi-LED-Illumination-and-Imaging) [82]. For heliotropic bending analysis in the field, time-lapse images were taken using BirdCam 2.0 (Wingscapes) cameras. Stem angles from the base of the first internode to the apex were measured using ImageJ software [83].

## RNA sequencing analysis

For all RNA-seq experiments, stem peels were collected from the first internodes of 2-week old plants using a vegetable peeler. Peels were collected from just below the stem apex to the bottom of the first internode. Consistent pressure was used for each peel to ensure similar thickness for each sample. The entire peel from each plant was frozen in liquid nitrogen and homogenized for use in extractions. For the field RNA-seq experiment, samples were collected at 4-hour intervals over 48 hours, starting at dawn on August 18, 2016. Day length was 13.5 hours; temperatures during this period are shown in S11A Fig. Tissue from the east and west sides of stems from 6 plants was collected at every time point. Two peels were pooled before RNA extraction to generate 3 biological replicates per time point. For the phototropism experiment, samples were collected from the lit and shaded sides of stems at ZT3, ZT5, ZT7, ZT9, ZT12, ZT15, ZT18, and ZT21. Stem peels from 2 different plants were pooled for each biological replicate, for a total of 6 biological replicates for each side at each time point. RNA was extracted and libraries were built using a previously described high-throughput RNA library preparation protocol [84]. Libraries from each experiment were separately multiplexed and sequenced at the California Institute for Quantitative Biosciences (QB3) using HI-seq4000 and 50-base pair single-end runs.

The sunflower transcriptome was built using the Rpackage cufflinks v2.2.1 gffRead using the Ha412HOv2.0 genome and the Eugene annotation (HAN412_Eugene_curated_v1) [29,85]. Reads were quality filtered using the fastx toolkit quality filter to have a minimum quality score of 20 in at least 95% of bases [86]. Reads counts were generated using salmon-1.4.0 to align and quantify the reads against the generated transcriptome [87]. The field data had an average of 16.8 million counts per sample, and the phototropism data had an average of 12.1 million counts per sample. Genes were removed if they did not have at least 10 counts in at least 3 samples for the field experiment, or in at least 6 samples in the phototropism experiment. Reads were normalized using the R package edgeR with the TMM method, and differentially expressed genes were determined by pairwise comparisons of sides of the stem (east versus west or shaded versus lit) at each time point, FDR < 0.05 [88]. For the differential expression analysis of the field data, the 2 days were combined into one, so each time point had 6 replicates per side. Rapidly induced genes during phototropism were identified by the overlap of the genes more highly expressed on the shaded side at ZT4 over ZT3, and ZT4

shaded over ZT4 lit during phototropism. Lit late induced genes during phototropism were identified by the overlap of those more highly expressed on the lit side at ZT15 over the shaded size at ZT15, and ZT18 lit over ZT9 lit during phototropism (FDR <0.05).

GO enrichment was performed using the R package clusterProfiler [89]. Arabidopsis homologs of each sunflower gene were found using blastx against the Arabidopsis proteome (TAIR10_pep_20101214), with an evalue cutoff of $< 9.9 \times 10^{-10}$ (S11 Data). GO terms were then assigned to each sunflower gene based on the top Arabidopsis hit using the terms found in ATH_GO_GOSLIM.txt.gz. Enriched GO terms were then found for each list of differentially expressed genes using all expressed genes as the background, FDR < 0.05.

Diel expression patterns of the field edgeR normalized counts were analyzed using the R package DiscoRhythm [35]. The Cosinor method was used to calculate acrophase, amplitude, and rhythmicity (FDR < 0.05) for each gene on both sides of the stem. Genes with an acrophase difference of >3 hours were considered differentially phased across the stem. Genes with >2-fold differences in amplitude across the stem were considered to have different amplitudes. PCA was performed on the heliotropism edgeR log transformed normalized counts using the R function prcomp [33].

## qPCR analysis

Sunflowers at 2 weeks after germination were transplanted into the field after sunset. Stem tissue was collected from the east and west sides of plants, starting at dawn, every 2 hours for 1 day and every 4 hours for 2 more days. Tissue was collected from 6 plants for each time point, and 2 plants were pooled for each biological replicate for a total of 3 biological replicates per time point. RNA was extracted using Spectrum plant total RNA kit (Sigma), and cDNA was synthesized from 300 ng of total RNA using Superscript III reverse transcriptase (Invitrogen). Quantitative PCR was performed using BioRad CFX96 as previously described [90]. Expression was normalized to 2 house keeping genes, *ELONGATION FACTOR 1-α* (*Ha412HOChr11g0495951*) and *UBIQUITIN-CONJUGATING ENZYME 21* (*Ha412HOChr06g0274091*), and analyzed using BioRad CFX Manager 3.1 software. Statistical significance was calculated using Welch's *t* test, pvalue < 0.05. Primers used for qPCR are found in S12 Data.

## Filter box assays

Experiments with filter boxes were conducted between August 12, 2021 and September 17, 2021, in Davis, California; day lengths during this period ranged between 13 hours 37 minutes and 12 hours 21 minutes. Mean, maximum, and minimum temperatures during these trials are shown in S11B Fig. Filter boxes were constructed with Rosco Roscolux light filters (Stage Lighting Store, Florida), #00 clear, #15 deep straw, and #375 cerulean blue. Light filters were wrapped and secured around garden support stakes to form a large box. A space of 1 to 2 inches was left at the top of the box between the walls and roof to provide air flow, and awnings of the appropriate filters were added to prevent nonfiltered light from entering through the openings. The boxes were placed into the field, held in place by garden stakes driven into the dirt. Light quality measurements were taken using a Black Comet spectrophotometer (StellarNet, Florida) from inside each of the boxes at the position and height of the plants. When measured at 11:30 AM, total PAR (400 nm to 700 nm) was 1,363 μmol m$^{-2}$ s$^{-1}$ within the clear box, 821 μmol m$^{-2}$ s$^{-1}$ within the deep straw box, and 410 μmol m$^{-2}$ s$^{-1}$ within the cerulean blue box. Approximately 4 to 6 plants were transplanted and monitored in each box for 3 to 4 days per trial. Mean and minimum and maximum daily temperatures during each of the 5 trials are shown in S11 Fig.

Rhythmicity of stem angle measurements was analyzed for each plant using the Biodare2 [91] eJTK test method with a pvalue < 0.001, $n = 22$. Percent stem growth was calculated by measuring the length from the bottom of the first internode to just below the apex, from dawn on the first day until dawn of day 5. Statistical significance was calculated using Welch's $t$ test, pvalue < 0.05.

## Supporting information

**S1 Fig. Cross-section of sunflower stem and stem peels. (A)** Cross-section of whole sunflower stem. Stem section stained with 0.05% toluidine blue and image taken at 25× magnification using a dissection scope. **(B)** Cross-section of stem peel stained with 0.01% toluidine blue and image taken at 100× magnification using a Zeiss Axioskop 2 plus.
(TIFF)

**S2 Fig. GO enrichment analysis of differentially expressed genes during phototropism.** GO enrichment analysis of genes differentially expressed genes during phototropism between ZT4 and ZT21. All terms for processes involved in light signaling, hormone regulation, and growth are shown for lit and shaded sides of stems. The underlying raw data may be found in S2 Data.
(TIFF)

**S3 Fig. GO enrichment analysis of differentially expressed genes during phototropism.** GO enrichment analysis of genes differentially expressed genes during phototropism between ZT4 and ZT21. All terms for processes involved hormone regulation, growth, cell wall–related processes, translation, and ribosomal processes are shown for lit and shaded sides of stems. The underlying raw data may be found in S2 Data.
(TIFF)

**S4 Fig. Overlap of up-regulated genes during phototropism and autostraightening. (A)** Venn diagram of all up-regulated genes on the shaded side at ZT4 and lit side at ZT15 in phototropic plants. **(B)** Venn diagram of genes assigned the "Response to Auxin" GO term that are up-regulated on the shaded sides of stems at ZT4 and on the lit sides of stems at ZT15 in phototropic plants. The underlying raw data may be found in S3 Data.
(TIFF)

**S5 Fig. Expression of core clock gene homologs during heliotropism. (A)** *RVE8*-like (Ha412HOChr07g0311811). **(B)** *RVE8*-like (Ha412HOChr14g0679651). **(C)** *RVE6*-like (Ha412HOChr16g0792081). **(D)** *RVE6*-like (Ha412HOChr09g0395701). **(E)** *PRR7*-like (Ha412HOChr08g0341571). **(F)** *PRR7*-like (Ha412HOChr01g0042171). **(G)** *PRR3*-like (Ha412HOChr14g0684881). **(H)** *PRR3*-like (Ha412HOChr07g0319851). **(I)** *TOC1*-like (Ha412HOChr02g0065721). **(J)** *LUX*-like (Ha412HOChr08g0371471). **(K)** *ELF3*-like (Ha412HOChr07g0293611). **(L)** *ELF3*-like (Ha412HOChr02g0074131). **(A-L)** Normalized expression over time for stems undergoing heliotropism (mean +/− SEM, $n = 3$). The underlying raw data may be found at NCBI GEO, accession GSE229654.
(TIFF)

**S6 Fig. Expression of genes with different diel phases across heliotropic stems during phototropism. (A, B)** Genes with different acrophases on the east and west sides of stems; west side acrophases between ZT16 and ZT24 **(A)** or between ZT8 and ZT16 **(B)**. **(C, D)** Zscore-normalized mean expression of these differentially phased genes (shown in **A, B**) during phototropism. Lines fitted with LOESS. Ribbons represent 95% confidence intervals. The underlying raw data may be found in S6 Data for S6A and S6B Fig and in S17 Data for S6C

and S6D Fig.
(TIFF)

**S7 Fig. Genes with amplitude differences in expression during heliotropism are not correlated with initiation of phototropism.** (**A**) Genes with a >2-fold difference in amplitude between east and west sides of stems during heliotropism. Zscore-normalized mean CPM of these genes during phototropism. Lines fitted with LOESS. Ribbons represent 95% confidence intervals. (**B**) Venn diagram showing overlap between those genes with an amplitude difference during heliotropism and genes with a phase difference during heliotropism. The underlying raw data may be found at NCBI GEO, accession GSE229654 and in S7 and S17 Data for S7A Fig; and in S6 and S7 Data for S7B Fig.
(TIFF)

**S8 Fig. Differential gene regulation between phototropism, onset of heliotropism, and established heliotropism.** (**A**) *IAA4*-like (*Ha412HOChr10g0435441*). (**B**) *SAUR12*-like (*Ha412HOChr07g0303111*). (**C**) *IAA4*-like (*Ha412HOChr01g0004341*). (**D**) *PIN7*-like (*Ha412HOChr01g0035481*). (**E**) *PHYB*-like (*Ha412HOChr02g0079361*). (**F**) *HB2*-like (*Ha412HOChr13g0623611*). (**G**) *BIM1*-like (*Ha412HOChr02g0064141*). (**H**) *KAT1*-like (*Ha412HOChr04g0179931*). (**I**) *EIN3*-like (*Ha412HOChr15g0726161*). (**J**) *EBF1*-like (*Ha412HOChr01g0047431*). (**K**) *NPY8*-like (*Ha412HOChr12g0580251*). The left graph in all panels is the normalized expression over time on shaded and lit sides of stems (mean +/− SEM, *n* = 6) undergoing phototropism in a growth chamber. The middle graph for all panels is normalized expression during the first 3 days in the field (mean +/− SEM, *n* = 3, except ZT64 west, *n* = 2). Statistical significance calculated using Welch's *t* test, qvalue < 0.05. The right graph for each panel is normalized expression over time for stems maintaining heliotropic movements (mean +/− SEM, *n* = 3). The underlying raw data may be found in S14 Data for the middle plots, and at NCBI GEO, accession GSE229654, for the plots on left and right sides of the figure.
(TIFF)

**S9 Fig. Photos of light filter boxes.** Photos taken in the field of light filter boxes, (**A**) clear box, (**B**) blue-depleting box, (**C**) red-depleting box.
(TIFF)

**S10 Fig. Comparison of sunflower phototropism and heliotropism.** (**A**) Sunflower plants bending towards blue light in a growth chamber show transcriptional responses typical of the phototropin signaling pathway. (**B**) Daytime heliotropic movements likely depend on multiple types of photoreceptors but do not display transcriptional patterns associated with specific hormone signaling pathways. Nighttime heliotropic movements are associated with extensive differential gene expression across the stem, with enrichment for auxin signaling and response genes on the faster growing, west sides of stems.
(TIFF)

**S11 Fig. Temperatures during field trials.** (**A**) Temperatures during collection of tissue samples for RNA-seq analysis of gene expression during heliotropism. (**B**) Average temperatures during each filter box trial. Ribbons represent the minimum and maximum temperatures during each trial. All temperatures are in degrees Celsius. The underlying raw data may be found in S16 Data.
(TIFF)

**S1 Table. Statistics for overlap of differentially expressed genes.** Fisher's exact test run on each combination of the overlap of differentially expressed genes as shown in Fig 3M. (TIFF)

**S1 Data. Differential gene expression of the shaded and lit sides of sunflower stems during phototropism.** Pairwise comparisons of the normalized counts of the shaded and lit sides of sunflower stems at each time point collected. Genes with a logFC of > 0 are more highly expressed on the shaded side and genes with a logFC of < 0 are more highly expressed on the lit side. (XLSX)

**S2 Data. Gene ontology enrichment analysis of differentially expressed genes during phototropism.** All statistically significantly enriched GO terms for genes significantly more highly expressed on one side of the stem at each time point. (XLSX)

**S3 Data. List of differentially expressed genes shown in S2 Fig.** List of genes more highly expressed on the shaded side of phototropic stems at ZT4, the lit side at ZT15, and their overlap. List of genes overexpressed at ZT4 shaded and ZT15 lit, which are found in the response to auxin category. (XLSX)

**S4 Data. Diel analysis of the heliotropism transcriptome.** Analysis of rhythmic expression of genes on either the east or west sides of stems during 2 days of heliotropism. (XLSX)

**S5 Data. Gene ontology enrichment analysis of genes with different peak phases during heliotropism.** All statistically significantly enriched GO terms for genes with diel rhythms. Each gene was assigned to one of six 4-hour bins. (XLSX)

**S6 Data. List of genes with a phase difference during heliotropism.** All sunflower genes with a greater than 3-hour acrophase difference between the east and west sides of stems during heliotropism. (XLSX)

**S7 Data. List of genes with an amplitude difference during heliotropism.** All sunflower genes with an amplitude fold change greater than 2 between the east and west sides of stems during heliotropism. (XLSX)

**S8 Data. Differential gene expression across the east and west sides of sunflower stems during heliotropism.** Pairwise comparisons of the normalized counts of the east and west sides of sunflower stems at each time point. Genes with a logFC of > 0 are more highly expressed on the west side and genes with a logFC of < 0 are more highly expressed on the east side. (XLSX)

**S9 Data. Gene ontology enrichment analysis of genes differentially expressed during heliotropism.** All statistically significantly enriched GO terms for genes with differential expression across the stem at each time point. (XLSX)

**S10 Data. Gene ontology enrichment analysis of genes with different phases across the stem during heliotropism.** All statistically significantly enriched GO terms for rhythmic

genes with a greater than a 3-hour difference in acrophase across the stem during heliotropism. Genes were split into early day-phased or night-phased groups.
(XLSX)

**S11 Data. Annotation file including sunflower gene IDs and Arabidopsis homologs.** Arabidopsis homologs of each sunflower gene were found using blastx against the Arabidopsis proteome (TAIR10_pep_20101214), with an evalue cutoff of $< 9.9 \times 10^{-10}$. The single best Arabidopsis hit was retained for each gene search.
(XLSX)

**S12 Data. Primer sequences used for qPCR.**
(XLSX)

**S13 Data. Stem angle measurements.** Angles of plant undergoing phototropism or heliotropism; data are plotted in Figs 1B, 1C, 2B, 4A and 4J.
(XLSX)

**S14 Data. Gene expression levels as determined by qPCR.** Normalized data plotted in Figs 4B–4H and S8 are provided, as are the Cq values of the independent biological replicates for both the genes of interest and housekeeping genes.
(XLSX)

**S15 Data. Light spectra and analysis of heliotropism in altered light conditions.** Spectral data plotted in Fig 4I as well as circadian parameters for heliotropic movement of plants grown in these conditions (Fig 4K and 4L) are provided.
(XLSX)

**S16 Data. Temperature conditions during field heliotropism experiments.** Temperatures when stem samples were collected for RNA-seq and during heliotropism under altered light conditions; data are plotted in S11 Fig.
(XLSX)

**S17 Data. Expression data for genes with mean expression values plotted in Figs 3, S6 and S7.** Z-score-normalized expression data for genes with mean values plotted in Figs 3D, 3G, 3H, 3K, 3L, S6C, S6D and S7A. The raw and counts per million normalized counts for these transcripts may be accessed at NCBI GEO, accession # GSE229654.
(XLSX)

**S18 Data. Gene ontology enrichment analysis of genes differentially expressed across stems during both phototropism and heliotropism.** Statistically significantly enriched GO terms for genes differentially expressed across stems in the comparisons shown in Fig 3M and 3N.
(XLSX)

## Acknowledgments

We thank Nicky Creux and Carine Marshall for their assistance in collecting tissue and running experiments. We also thank Julin Maloof and John Davis for advice on transcriptome analysis.

## Author Contributions

**Conceptualization:** Christopher J. Brooks, Hagop S. Atamian, Stacey L. Harmer.

**Formal analysis:** Christopher J. Brooks.

**Funding acquisition:** Stacey L. Harmer.

**Investigation:** Christopher J. Brooks, Hagop S. Atamian.

**Supervision:** Stacey L. Harmer.

**Writing – original draft:** Christopher J. Brooks.

**Writing – review & editing:** Christopher J. Brooks, Hagop S. Atamian, Stacey L. Harmer.

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
