## [Editor Report · Decision Letter 0]

24 Apr 2023

Dear Stacey, 

Thank you for submitting your manuscript entitled "Transcriptome and physiological analyses implicate multiple light signaling pathways in the control of solar tracking in sunflower" for consideration as a Discovery Report by PLOS Biology.

Your manuscript has now been evaluated by the PLOS Biology editorial staff as well as by an academic editor with relevant expertise and I am writing to let you know that we would like to send your submission out for external peer review.

Once your full submission is complete, your paper will undergo a series of checks in preparation for peer review. After your manuscript has passed the checks it will be sent out for review. To provide the metadata for your submission, please Login to Editorial Manager (https://www.editorialmanager.com/pbiology) within two working days, i.e. by Apr 26 2023 11:59PM.

Kind regards,

Ines

--

Ines Alvarez-Garcia, PhD

Senior Editor

PLOS Biology

---

## [Decision Letter · Decision Letter 1]

28 Jun 2023

Dear Stacey,

Thank you for your patience while your manuscript entitled "Transcriptome and physiological analyses implicate multiple light signaling pathways in the control of solar tracking in sunflower" went through peer-review at PLOS Biology as a Discovery Report. Please also accept again my apologies for the delay in providing you with our decision. Your manuscript has now been evaluated by the PLOS Biology editors, an Academic Editor with relevant expertise, and by two independent reviewers.

As you will see, both reviewers find the conclusions novel and significant for the field, but they also raise several issues that would need to be addressed. These mainly include clarifications and suggestions to improve the text and structure of the manuscript to make it more accessible for readers. In addition, they suggest some analyses to make the best out of the dataset, but after consulting with the Academic Editor, we have decided not to make these a requirement for publication.

In light of the reviews, we are pleased to offer you the opportunity to address the comments from the reviewers in a revision that we anticipate should not take you very long. We will then assess your revised manuscript and your response to the reviewers' comments with our Academic Editor aiming to avoid further rounds of peer-review, although might need to consult with the reviewers, depending on the nature of the revisions.

**IMPORTANT - SUBMITTING YOUR REVISION**

3. Resubmission Checklist

a) *PLOS Data Policy*

b) *Published Peer Review*

Sincerely,

Ines

--

Ines Alvarez-Garcia, PhD

Senior Editor

PLOS Biology

Reviewers' comments

Rev. 1:

In this manuscript by Brooks et al., the authors study how sunflower plants track the sun as it moves across the sky each day, a process known as heliotropism. Previously, it was predicted that sunflower heliotropism is a specialized form of phototropism, the bending of plants towards light, but this has not been shown empirically. They use two main techniques to address this question, physiological measurements of stem bending and transcriptomics. They grow sunflower plants in a phototropic light regime in which blue light is held in place to trigger phototropic response. Then they grow plants in heliotropic conditions, including outdoor grown sunflower exhibiting heliotropic movements and indoor grown sunflower plants not exhibiting heliotropic movements but then transferred outdoors where they begin heliotropic movement. They measure stem angles under each of these conditions and collect timecourse RNA-seq samples from stem sections that are either "shaded" (pointed away from the light source) or "lit" (pointed toward a light source). They then use the RNA-seq to track transcriptional changes and compare these changes between plants undergoing phototropism to plants undergoing heliotropism. From these experiments, they find that heliotropism and phototropism show different transcriptional changes. Most prominently, they show that changes in genes responsive to auxin are a core part of phototropic responses but are not a prominent part of heliotropism during the day, only at night. Instead, heliotropism shows changes in genes related to shade avoidance. This prompted them to grow sunflower plants outdoors in heliotropic conditions with either blue or red light filtered out of their environment, and they found that heliotropism continued under either of these conditions. From this work they conclude that heliotropism is not simply a specialized form of phototropism, but rather it constitutes a different type of directional light response that relies on a combination of light signaling pathways.

This work addresses an important, unanswered, and technically challenging question in the field of plant rhythms, what cellular systems do plants use to track the sun as it moves across the sky? Because most model species do not have obvious heliotropic movements, the authors use sunflower plants, which have heliotropic movement but lack genetic tools. Thus, they use transcriptomics and generate an impressive dataset to establish some fundamental ideas about heliotropic movement. I have no major concerns with the approach. I do have a few concerns about the analyses and whether there are additional comparisons that might reveal more about the nature of heliotropism. I also have some suggested ways to improve the clarity of the manuscript. These are outlined below.

1. I had a bit of trouble following the logic of the paper since there were many dimensions of data that were being explained (i.e. time, stem illuminance, growth condition). I wonder if there is a way to create some simple diagrams or models that can be included in the main figures that make the experimental design more readily accessible to the reader? At the end one could create a summary model that incorporates the results with the diagrams demonstrating the larger conceptual idea.

2. I think overall there could be increased clarity in the written presentation of the data. This might include clearer justifications for the experiments at the beginning of each section. For instance, I had a bit of trouble understanding why the autostraightening was important to the manuscript. This is likely my own lack of knowledge, but there could be bit more conceptual justification. My thought is that autostraightening could potentially explain some of the heliotropic effect, but that was not clearly stated. I think increased justification and explanation would help with each of the large experiments presented in the manuscript.

3. Very large expression datasets from plants collected at multiple timepoints were generated here, but the data seem to be underutilized in the comparative analyses. It was unclear why only certain timepoints were chosen for comparisons. For instance, only ZT4 and ZT15 were chosen for the phototropism data when compared to the heliotropic data. Why wasn't there a more comprehensive comparison of the heliotropic data to more timepoints in the phototropic data? This may have revealed that other timepoints in the phototropic data matched the heliotropic data. In this sense, it seems like a lot of the data that was collected wasn't discussed in detail.

4. It was unclear why the ZT15 timepoint was chosen as the autostraightening point when comparing to heliotropism. From the phototropic stem bending figure, that seemed like a timepoint that would be too late to catch the initial gene expression changes that would have led to autostraightening. Please explain this in more detail, and if necessary, compare to other autostraightening timepoints.

Rev. 2:

In their manuscript the authors address the molecular basis of heliotropism, and provide with evidence that this response is different from phototropism, as genes rapidly responsive to phototropism, present only minor changes during heliotropism. Importantly data supports the involvement of not only blue light receptors, but potentially the Red/FR light phytochromes in solar tracking, showing that heliotropism is not a specialized of phototropism dependent on phot-blue light receptors. This is an important basepoint to dissect the complexity of differential growth responses where circadian and light signals interact.

The transcriptomic study conducted is extensive, and most conclusions supported by the study. However, some comments and suggested adjustments follow.

Material and Methods.

1. Figure S1 illustrates the cross sections of the stems and stem peels used. The figure itself needs indication of the main structures shown. It is also not clear in the material and methods, how the use of the peels led to the dissection of the west and east samples used in the study. How did the dissection happened?, what was the criteria at the cellular level to decide on sampling and ensure samples here as homogeneous as possible? was a criteria of cell size applied in addition to location of the cell? or?

2. Can the authors include in the filter box assays the other environmental factors that impact on growth responses (i.e temperature, PAR differences, photoperiodic light/dark variation).

Results/Discussion

1. Figure 1D. In their analysis of phototropism and genes upregulated in the shaded side of stems, they mention Auxin genes were as expected prevalent, but also genes involved in GA and BR sensing/signaling. BZR1 is in Fig 1D provided as example, but can the authors provide with a table of the rest of the hormonal pathways related genes ?

Same for the auto-straightening response. How many and which gene homologues of auxin genes were identified? (Fig 1E, 1F).

2. Figure 2D, 2E and S1 data. The authors mention in the text that homologues of clock genes displayed strong daily rhythms, and provide the profiles for LHY and PRR5. Probably it would be of interest to include the rest of the clock genes singled out in the analysis ( as part of the S1 data or in a linked table?).

3. Figure S2 Can the authors provide in Supplementary Tables the gene list related to this figure?

4. For the field-experiment with grown sun flower. and expression of genes in the east and west side of the stem. Can the authors comment on the other environmental conditions that applied to the experiment (i.e temperature, dark/light hours, light quality spectra) PAR, R:FR ratio.

5. For the same experiment, While the authors describe some of the enriched functions of the genes identified in the afternoon and evening (ZT8-12) (ZT12-16),some of the enriched processes like cell division (73 genes) and RNA modifications (|T12-16- 145 genes) as well as BR signaling (46 in ZT20-24) or Defense genes (83), root development were also enriched. While the last two categories may not be directly linked, can the authors include a comment on the other enriched ones that may have a link or lead to novel unexplored areas?

6. In the transcriptomic analysis of auxin signaling in heliotropic growth. Can the authors include in supplementary tables the 1369 genes that have acrophase difference greater than 3 hours , those that have different wave forms on the two sides (Fig 2J) ad the 811 with two fold difference in amplitude across the stem?

7. The authors mention that in their solar tracking experiment, they found enrichment of blue light and UV-responses on the east sides of stems in the morning and evening. A list of the genes identified would be desirable to include in Supplementary tables.

8. In the same solar tracking experiment, in ZT12-16, there seems to be enrichment of ribosomal proteins and translation. A comment on the potential relevance of these findings in terms of novel mechanisms to explore in terms translation links to the joint mechanisms for differential growth by the circadian clock and light would be desirable to include in discussion.

9. In the transcriptional responses to phototropism and heliotropism section. Fig 2G singles out the S15a-like gene, but it may be more suitable to include as examples of genes that could be related to the differential growth responses.

10. Interestingly the authors report that hundred of genes with different amplitudes of expression on opposite sides of heliotropic stems did not show significantly enriched functional categories, and there was limited overlap between genes expressed within different phases and amplitudes in the field. Can the authors comment on the potential impact on other environmental factors (i.e temperature, light intensity) on these set of genes?. How much variation linked to other environmental factors could be present in this study between CE studies of phototropism and the heliotropic response?

11. It is very interesting the identification of phytochrome responses as part of the heliotropic response and the initiation of solar tracking. Would it be possible to extend on the potential impact of the phytochromes in the response, using available genomic datasets in terms of genes responsiveness to R:FR ratios, R-intensity?, also is the potential impact of phy on these genes affecting the phase or affecting the amplitude?

12. Can the authors comment in the manuscript on the potential differences brought by the light receptors into the growth responses?. They provide with evidence that the onset and maintenance of the heliotropic response can tolerate light quality. Based on what we know about the photoreceptors and their dynamic control of growth, can the authors extend on the potential avenues and mechanisms behind the role of the light quality receptors and day/night differences in growth?

13. Could it be possible to map across the heliotropic responsive genes, with differences in phases and/or amplitude what is the potential contribution of the phys, UVR8 and phots? (even if for selected examples and relating to data across other plant species).

14. In the discussion, it is clearly stated that the focus of the study is transcription but that other mechanisms may be involved in the heliotropic response. Probably worth including together with post-translational cascades, translation (as Fig S7 shows enrichment for some translation related processes).

General comment. It would be desirable to include annotation in all the Supplementary data tables where gene lists are presented.

---

## [Editor Report · Decision Letter 2]

18 Aug 2023

Dear Stacey,

Thank you for your patience while we considered your revised manuscript entitled "Transcriptome and physiological analyses implicate multiple light signaling pathways in the control of solar tracking in sunflower" for publication as a Discovery Report at PLOS Biology. This revised version of your manuscript has been evaluated by the PLOS Biology editors and the Academic Editor.

Based on our Academic Editor's assessment of your revision, we are likely to accept this manuscript for publication, provided you satisfactorily address the data and other policy-related requests stated below.

In addition, we would like you to consider a couple of suggestions to improve the title:

"Multiple light signaling pathways control solar tracking in sunflowers"

or (if that one is too strong)

"Regulation of solar tracking in sunflowers differs from phototropism control and involves multiple light signaling pathways 

We expect to receive your revised manuscript within two weeks. 

*Published Peer Review History*

*Press*

Sincerely,

Ines

--

Ines Alvarez-Garcia, PhD

Senior Editor

PLOS Biology

DATA POLICY:

I note you have submitted several data files, but it's unclear to what figures they belong. We ask that all individual quantitative observations that underlie the graphs shown in the figures are made available in one of the following forms:

Fig. 1B-G; Fig. 2A-C, E-M; Fig. 3A-L, N; Fig. 4A-L; Fig. S2; Fig. S3; Fig. S5A-L; Fig. S6A-D; Fig. S7A; Fig. S8A-K and Fig. S11A, B

Please also ensure that figure legends in your manuscript include information on WHERE THE UNDERLYING DATA CAN BE FOUND.

**Please also make available at this stage the data deposited in GEO (ID: GSE229654).

---

## [Editor Report · Decision Letter 3]

21 Sep 2023

Dear Stacey,

Thank you for the submission of your revised Discovery Report entitled "Multiple light signaling pathways control solar tracking in sunflowers" for publication in PLOS Biology. On behalf of my colleagues and the Academic Editor, Mark Estelle, I am delighted to let you know that we can in principle accept your manuscript for publication, provided you address any remaining formatting and reporting issues. These will be detailed in an email you should receive within 2-3 business days from our colleagues in the journal operations team; no action is required from you until then. Please note that we will not be able to formally accept your manuscript and schedule it for publication until you have completed any requested changes.

PRESS

Sincerely, 

Ines

--

Ines Alvarez-Garcia, PhD

Senior Editor

PLOS Biology
